# The Effect of Ferulic Acid-Grafted Chitosan (FA-g-CS) on the Transmembrane Transport of Anthocyanins by *sGLT1* and *GLUT2*

**DOI:** 10.3390/foods11203299

**Published:** 2022-10-21

**Authors:** Yi Ma, Xiaojiao Chen, Tiwei Diao, Yinjiang Leng, Xiaoqin Lai, Xin Wei

**Affiliations:** College of Biological Engineering, Sichuan University of Science and Engineering, 188 University Town, Yibin 644000, China

**Keywords:** anthocyanins, transmembrane transport, *sGLT1*, *GLUT2*, *siRNA*

## Abstract

This work aims to evaluate the effect of ferulic acid-grafted chitosan (FA-g-CS) on the interaction between anthocyanin (ANC) and *sGLT1*/*GLUT2* and their functions in ANC transmembrane transport using Caco-2 cells. The transmembrane transport experiments of ANC showed its low transport efficiency (Papp < 10^−6^ cm/s), whereas the phenomenon of a significantly rise in anthocyanins transport efficiency was observed with the incubation of FA-g-CS (*p* < 0.05). In order to investigate the mechanism of FA-g-CS improving ANC transmembrane transport, Caco-2 cells were transfected with small interfering RNA (*siRNA*) specific for transporters *sGLT1* and *GLUT2*, and incubated with ANC, FA-g-CS, or their combination. Subsequently, Western blot analyses and immunofluorescence staining were carried out to monitor the intracellular *sGLT1* and *GLUT2* levels. These siRNA-transfected cells, incubated with compounds, indicate that *sGLT1* and *GLUT2* participated in the ANC transmembrane transport and that FA-g-CS, ANC, or their combination enhance *sGLT1*/*GLUT2* expression. In particular, Caco-2 cells incubated with both FA-g-CS and ANC show significantly increased *sGLT1* or *GLUT2* expression (>80%) compared with exclusively using FA-g-CS or ANC (<60%). Molecular docking results demonstrate that there is a good binding between FA-g-CS/ANC and *sGLT1* or *GLUT2*. These results highlight that FA-g-CS promotes the transmembrane transport of ANC by influencing the interaction between ANC and *sGLT1*/*GLUT2*; the interaction between FA-g-CS and ANC could be another key factor that improves the bioavailability of ANC.

## 1. Introduction

Anthocyanins (ANCs) are a kind of natural pigment in the flavonoid family with a plethora of biological activities [1,2]. Previous reports have indicated that ANCs possess great potential in preventing cardiovascular disease, tumors, and obesity [3]. Although they exhibit interesting bioactivity characteristics, the efficacy of ANCs is limited by their low bioavailability [4]. It has been indicated that the limited absorption [5] of ANCs is associated with the oxonium ion next to C2, which renders ANCs highly vulnerable to nucleophilic attack by compounds such as hydrogen peroxide, water, and ascorbic acid.

Various attempts (such as acylation, thermal processing, etc.) have been adopted to improve the stability and antioxidant activity of ANCs [6]. Among them, microencapsulation technology is an important approach for enhancing the stability of ANCs [7,8]. Several embedding technologies, such as solid-phase particles, microemulsions, and liposomes, have shown promising results in improving the stability of ANCs [9]. Although these approaches prevent ANC degradation and increase the gastrointestinal digestion residue, the bioavailability of ANCs was not substantially improved due to their transmembrane transport [10]. Evidence suggests that ANCs mainly permeate the intestinal mucosa through transport mediated by *sGLT1* and *GLUT2* glucose transporters [11].

The interactions between food components and transporters are among the factors that might affect the transmembrane transport of ANCs. The inhibition of glucose absorption due to the interaction of ANCs and *sGLT1* as well as *GLUT2* has been well-studied [11,12,13]. Genetic and chemical inhibitors of *sGLT1* and *GLUT2* were used to verify the extent to which ANCs depend on them for intestinal absorption, which showed that *sGLT1* interacts with a greater range of ANCs, while *GLUT2* is the main transporter for specific ANC glycosides [14]. Moreover, it is speculated that quercetin may compete with ANCs at the same binding site of *sGLT1* [15]. Narasimhan provided molecular biology and animal data proving that ferulic acid (FA) could regulate the expression of *GLUT2* [16]. In addition, the esterification products of monosaccharides and polysaccharides were able to regulate the activity of *GLUT2* [17].

Few studies have demonstrated the operation and mode of action of microcapsule components (the wall materials) and carriers. In our previous studies, the bioavailability of ANCs was noticeably improved by grafting FA onto polysaccharides for preparing ANC microcapsules [18,19]. In addition, epigallocatechin gallate nanoencapsulated in ferulic acid-grafted chitosan (FA-g-CS) was confirmed to show dramatically increased stability and antioxidant activity [20]. FA was extracted from plants with strong antioxidant activity. CS, with excellent biological property [21], has been widely used as an excellent nanocomplex vehicle to increase the bioavailability of some useful substances. Additionally, the interaction between ligands and FA has already been confirmed, and the conformation of these molecules may be changed by noncovalent interactions [22]. However, the exact mechanism of interaction between FA-g-CS and ANCs and its effect on antioxidation activity are not yet clarified. It is speculated that FA-g-CS may influence the interaction of anthocyanin and *sGLT1*/*GLUT2*, which promotes transmembrane transport of anthocyanins. Clarifying the stability mechanism of bioactive substances from the perspective of their interactions is a contemporary issue in the field of food and drug research [23].

The aim of the present study is to highlight the impact of FA-g-CS on the expression of *sGLT1*/*GLUT2* and its potential toxicity on Caco-2 cells, with emphasis on highlighting the mechanisms involved in the transmembrane transport of ANC. In particular, the interactions between FA-g-CS and ANC and between *sGLT1*/*GLUT2* and FA-g-CS or ANC are discussed to further understand the role of FA-g-CS in the transmembrane transport of ANC. This study may offer new ideas and methods toward the formulation of ANC microcapsules with high bioavailability.

## 2. Materials and Methods

### 2.1. Materials and Reagents

Cyanidin-3-glucoside was acquired from Ziguang (Nanjing, China, purity >98%). MEM was purchased from Procell Life Science and Technology Co., Ltd. (Wuhan, China). Ethanol (analytical-grade) was provided by Sinopharm Chemical Reagent Co., Ltd., Shanghai, China. MTT, DAPI, Triton X-100, and BSA were obtained from Sigma (St. Louis, MO, USA). PBS was obtained from Life Technologies (Thermo Fisher Scientific, Waltham, MA, USA) and FBS was obtained from HyClone (Waltham, MA, USA). Plastic dishes and plates were obtained from Corning, Inc. (Corning, NY, USA). RIPA lysis buffer, PMSF, BCA Protein Assay Kit, Cell Membrane Protein and Cytoplasmic Protein Extraction Kit, Nuclear and Cytoplasmic Protein Extraction Kit, Lactate Dehydrogenase (LDH) Cytotoxicity Test Kit, and goat serum were provided by Beyotime (Shanghai, China). SDS-PAGE Protein Loading Buffer 5X was obtained from AtaGenix (Wuhan, China). Skim milk powder was obtained from Inner Mongolia Yili Ind. Group Co., Ltd. (Inner Mongolia, China). ECL was purchased from Wuhan Juneng Yitong Biological Co., Ltd. (Wuhan, China). HRP to mark goat anti-rabbit secondary antibody and HRP to mark goat anti-mouse secondary antibody were purchased from Wuhan Sanying Biotechnology Co., Ltd. (Wuhan, China). Tris, glycine, and SDS were purchased from Solarbio (Beijing, China). Membrane nuclear and cytoplasmic protein extraction kits, 4% paraformaldehyde fixing solution, and RNase A solution were purchased from Sangon Biotech (Shanghai, China). MiniBEST Universal RNA Extraction Kit and PrimeScript RT Reagent Kit with gDNA Eraser (Perfect Real-Time) were purchased from Takara (Beijing, China). DEPC-treated water was purchased from Ambion (Carlsbad, CA, USA). Hepatocytes grown on glass coverslips were purchased from Nest (Wuxi, China), and 96-well plates were purchased from Costar (China). PI solution was provided by BioLegend (San Diego, CA, USA). CS had average molecular weight ~1.5 × 10^5^, degree of deacetylation ≥90.0%, and FA purity >98%. Preparation of FA-g-CS was conducted according to the previous study [24].

### 2.2. Transmembrane Transport Studies of ANC

In this work, we utilized and slightly modified a procedure from Zou et al. [11] to perform ANC transport studies. Firstly, ANC commercial powder was pre-dissolved in ultra-pure water and phosphate-buffered solution (PBS, pH7.4) was used for FA-g-CS dissolving. Then, pre-warmed phosphate-buffered saline (PBS) was used to wash and equilibrate the Transwell. Subsequently, the transport buffer containing ANC was added to the basolateral (1.5 mL) or the apical (0.5 mL) side of the inserts, while the corresponding volume of transport buffer was added to the receiving container. Samples measuring 50 uL were taken after 1, 2, 3, and 4 h incubation, and the same amount of pre-warmed PBS was replenished immediately. The samples were treated with HCl and frozen until HPLC analysis [11]. The ANC-absorption-enhancing studies with FA-g-CS across Caco-2 cells were conducted using a transport buffer containing FA-g-CS+ANC instead of ANC alone.

Transport efficiency (A→B (%)) was calculated according to the following formula: (concentrations of the compound at basolateral side overtime)/(concentrations of the compound at the apical side at zero hours) × 100. The apparent permeability coefficient (Papp) was defined as follows: Papp = (dQ/dt)(1/[AC_0_]); here, A represents the surface area of the Transwell, C_0_ represents the compound concentration added in the apical side, and the transport rate of the compound over time is denoted by dQ/dt.

### 2.3. Cell Culture and Viability Assays

Caco-2 cells were provided by the Cell Bank of the Chinese Academy of Sciences (Shanghai, China). MTT analysis was used to investigate cell viability [25]. In short, Caco-2 cells were incubated with ANC (0.23, 0.69, 2.06, 6.17, 18.52, 55.56, 166.67, or 500 × 10^−3^ mg/mL), FA-g-CS (0.09, 0.27, 0.82, 2.47, 7.41, 22.22, 66.67, or 200 × 10^−3^ mg/mL), or their respective combinations. Each group was incubated with three multiple wells for 48 h. Subsequently, the chemical medium was replaced with fresh MTT solution for 3 h at 37 °C and the crystal violet was dissolved with DMSO. Then, the optical density (OD) was measured at 570 nm using a PerkinElmer EnVision (PerkinElmer, Waltham, MA, USA). The following Formula (1) was used to calculate cell viability:(1)$$\mathrm{Cell}\;\mathrm{viability}\left(\%\right)=\frac{{\mathrm{OD}}_{\mathrm{sample}}-{\mathrm{OD}}_{\mathrm{blank}}}{{\mathrm{OD}}_{\mathrm{control}}-{\mathrm{OD}}_{\mathrm{blank}}}$$
where OD_control_, OD_sample_, and OD_blank_ represent the measurements of the control, sample, and blank, respectively.

### 2.4. Cell Cytotoxicity Assay

Due to cell membrane damage, the intracellular LDH was released into the culture medium. The LDH level was used as an indicator of irreversible cell death [26]. The method used here was based on the spectrophotometric determination at 490 nm of NADH disappearance.

Upon exposure to different concentrations of ANC, FA-g-CS, or their combinations, the culture medium was aspirated and centrifuged at 400× *g* for 5 min in tubes with a porous matrix to acquire a cell-free supernatant. A measure of 120 microliters of media was mixed with 60 μL of reagent in a 96-well plate, and the absorbance was recorded with PerkinElmer EnVision. The cytotoxicity of these substances was reflected using the following Equation (2):(2)$$\mathrm{LDH}\;\mathrm{activity}\left(\frac{\mathrm{mU}}{\mathrm{mL}}\right)=\frac{{\mathrm{OD}}_{\mathrm{sample}}-{\mathrm{OD}}_{\mathrm{blank}}}{{\mathrm{OD}}_{\mathrm{standard}}-{\mathrm{OD}}_{\mathrm{blank}\;\mathrm{of}\;\mathrm{standard}}}\times 1000\;\mathrm{mU}/\mathrm{mL}$$
where OD_sample_, OD_blank_, OD_standard_, and OD_blank_ represent the measurements of the sample, the blank, the standard, and the blank of standard, respectively.

### 2.5. RNA Interference

To examine the effect of *sGLT1* and *GLUT2* on ANC transmembrane transport in more detail, the Caco-2 cells were transfected with *siRNA* targeted to *sGLT1* and *GLUT2* (*sGLT1 siRNA-1*, *sGLT1 siRNA-2*, *sGLT1 siRNA-3*, *GLUT2 siRNA-1*, *GLUT2 siRNA-2*, and *GLUT2 siRNA-3*) to knockdown their expression on day 18 after differentiation. The specific experimental operation was carried out according to the literature [27]. *NC siRNA* was used as a negative control. The *siRNA* oligos and all transfection reagents were provided by Hippobio (Huzhou, China). Western blotting and real-time PCR (RT-PCR) analyses were performed to check the *sGLT1* and *GLUT2* expression levels after 72 h post-transfection. Table 1 shows the interfering RNA sequences.

### 2.6. RNA Extraction and RT-PCR

According to the manufacturer’s instructions, the MiniBEST Universal RNA Extraction Kit was used to isolate the total RNA from Caco-2 cells. Table 2 lists the three specific primers. GAPDH served as an internal control. RT-PCR was carried out using a CFX Connect Real-time PCR Detection System (Bio-Rad, Hercules, CA, USA) with cycling conditions for 10 min at 95 °C, followed by 40 cycles for 10 s at 95 °C, for 15 s at 60 °C, extension for 20 s at 72 °C, and annealing for 1 min at 65 °C. The 2^−ΔΔCt^ method [28] was adopted to calculate the relative quantification of the target gene.

### 2.7. Western Blot Assay

Firstly, the cells were washed twice with 1 mL of cold PBS and then lysed in 100 μL of ice-cold RIPA lysis buffer [29]. Western blot analysis was performed following the previous method with minor modifications [30]. Briefly, total protein samples were resolved by SDS-PAGE (8%, 10%, or 12%) and transferred to PVDF membranes blocked with 5% nonfat milk powder in TBST; these were then incubated with 8 μg of primary antibody at a dilution of 1:1000/500 (*v*/*v*) overnight at 4 °C. Detection was performed with 10 mL of ECL and oxidizing reagent and equal amounts of HRP-conjugated anti-mouse at a 1:5000 (*v*/*v*) dilution rate. To further illustrate the effect of FA-g-CS on the transmembrane transport of ANC, the Caco-2 cells were transfected with *siRNA* targeted to *sGLT1* and *GLUT2* (the optimal interference fragments screened by previous tests) and then incubated with FA-g-CS, ANC, or their combinations. Finally, the expressions of *sGLT1* and *GLUT2* were detected using a Western blot assay.

### 2.8. Immunofluorescence Staining

Immunofluorescence staining was carried out to further validate the effect of FA-g-CS/ANC on the expression of *sGLT1* and *GLUT2*. In short, Caco-2 cells were transfected with *NC siRNA* (negative control), *GLUT2-siRNA-2*, and *sGLT1-siRNA-3*; then, they were seeded onto a 4-well slide and chamber (Watson, Kobe, Japan). Subsequently, cells were incubated with 0.09 µg/mL FA-g-CS, 0.23 µg/mL ANC, or their combination at 37 °C for 48 h. Then, they were fixed with 4% paraformaldehyde for 10 min and soaked with 0.5% Triton X-100 in PBS for 15 min. In this study, goat serum was used to avoid nonspecific binding at room temperature. After 30 min of treatments, the cells were incubated with the primary antibody (β-actin mouse monoclonal (IgG1), Santa Cruz Biotechnology, with 1:100 dilutions) at 4 °C overnight [31], followed by incubation with the secondary antibody (goat anti-mouse IgG1-HRP, Santa Cruz Biotechnology, Shanghai, China, with 1:100 dilutions) for 1 h at 20–37 °C. Nuclear staining was performed with DAPI, and the images were captured with a fluorescence microscope (BX53, Olympus, Tokyo, Japan).

### 2.9. Molecular Docking

To further investigate the interactions between *sGLT1*, *GLUT2*, FA-g-CS+ANC, and ANC (Cyanidin-3-glucoside), we performed molecular docking analysis to identify their binding sites and to study the binding energy between the protein and the ligand. The two-dimensional structures of FA-g-CS and ANC were provided by the PubChem database (https://pubchem.ncbi.nlm.nih.gov/ accessed on 23 May 2022). Then, these two structures were energy-minimized and format-translated by ChemDraw 18.0 software (PerkinElmer, USA). The structures of FA-g-CS and ANC are shown in 3.7. Subsequently, pretreatment (hydrogenation, structural optimization, and energy minimization) was achieved using the Schrödinger software suite, which stores a database of ligand molecules for molecular docking.

The protein sequences of *sGLT1* and *GLUT2* were obtained using the Swiss-Model homology-modeling server (*sGLT1* template: 7FEN; GLUT2 template: 4ZWB). The homology model was built by the Protein Preparation Wizard available in the Schrödinger software suite, accessible through the Maestro 11.9 interface. Pretreatments involved removing the water from the crystal, repairing the bond information, minimizing the energy of the protein, and optimizing its geometric structure [32].

Molecular docking was conducted using the Glide module of Schrödinger Maestro software. All molecules were prepared based on the default setting of the LigPrep module of Schrödinger. For the results, the docking model with the lowest energy and bonding condition was determined to be the most favorable binding mode. Finally, the processes of molecular docking and screening were conducted by standard precision docking.

### 2.10. Statistical Analysis

In this work, all the results are expressed as the mean ± SD of three determinations. Statistical significance was tested at *p* < 0.05 by one-way analysis of variance (ANOVA) using IBM SPSS Statistics 20.

## 3. Results

### 3.1. Cell Viability

MTT analysis was carried out to identify the effect of ANC, FA-g-CS, and their combinations on the viability of Caco-2 cells. Figure 1 shows alterations in the viability of Caco-2 cells after 48 h of exposure to the prepared different concentrations of ANC and FA-g-CS, respectively. ANC with a concentration lower than 6.17 × 10^−3^ mg/mL showed no significant effect on cell viability (Figure 1A). Meanwhile, when the FA-g-CS concentration was <8.2 × 10^−4^ mg/mL, the cell viability was >90% (Figure 1B). The results also suggest that the combination of ANC (at <2.06 × 10^−3^ mg/mL) and FA-g-CS (at <0.82 × 10^−3^ mg/mL) had no noticeable influence on the viability of cells (Figure 1C).

### 3.2. Cell Cytotoxicity Assay

Figure 1D provides the results concerning the LDH assay. Up to 9 × 10^−5^ mg/mL of FA-g-CS showed a slight but insignificant influence on Caco-2 cells. In addition, ANC at the tested concentration (2.3 × 10^−4^ mg/mL) caused a negligible increase in cell cytotoxicity. Additionally, increasing the concentrations of FA-g-CS and ANC led to a significant effect on cell cytotoxicity over the 48 h experimental period. Interestingly, LDH activity increased with the concentrations of the additives, but no significant difference was observed between the relatively higher concentrations (2.7 × 10^−4^~2.47 × 10^−3^ mg/mL FA-g-CS). Meanwhile, concordant with the results obtained when FA-g-CS or ANC were added individually, for cells incubated with FA-g-CS and ANC, cytotoxicity was observed with higher concentrations of those supplements employed. Figure 1A–C show that the concentrations of 2.3 × 10^−4^ or 6.9 × 10^−4^ mg/mL ANC and 9 × 10^−5^ or 2.7 × 10^−4^ mg/mL FA-g-CS showed insignificant influence on Caco-2 cells. Considering the results of MTT and LDH analyses, a noncytotoxic concentration of ANC and FA-g-CS was used in follow-up experiments, specifically up to 9 × 10^−5^ mg/mL FA-g-CS and 2.3 × 10^−4^ mg/mL ANC.

### 3.3. Transport of ANC across Caco-2 Cell Monoplayers

The ANC transmembrane transport efficiency was measured using Caco-2 cells as a research model. As shown in Table 3, at a concentration of 5 × 10^−5^ or 1 × 10^−4^ mg/mL, the transport rate of ANC in the basolateral to apical side (B→A) was much lower than that from apical to basolateral (A→B). When ANC concentration reached 1.5 × 10^−4^ mg/mL, Papp(B→A)/Papp(A→B) was 1.29 and the transepithelial transport from A→B direction was saturated, which further suggests that transporters may be involved in ANC transmembrane transport [13]. Without FA-g-CS co-treatment, the overall ANC transport rate ranges from 0.81% to 1.18%, while the transport rate increased greatly with 5 × 10^−5^ mg/mL FA-g-CS addition (Table 3). Moreover, compared with the free-ANC group, the transport volume of ANC increased significantly within 1–4 h when ANC was combined with FA-g-CS (Table 4). In addition, FA-g-CS showed a significant effect on the transmembrane transport of higher concentrations of ANC (1 × 10^−4^, 1.5 × 10^−4^ and 2 × 10^−4^ mg/mL). Specifically, the transport volume of ANC after 4 h in FA-g-CS+ANC groups increased by 25%, 75%, 94%, and 74%, respectively, compared with the free-ANC group. The results suggested that FA-g-CS promoted the ANC transmembrane transport effectively.

### 3.4. Effect of siRNA-Mediated Knockdown of sGLT1 and GLUT2 in Caco-2 Cells

To gain deeper insights into the reliance of ANC uptake on *sGLT1* or *GLUT2*, the procedure found that Caco-2 cells treated with three different *siRNA* segments showed the *sGLT1* or *GLUT2* gene silencing phenomenon. The Western blot assay demonstrated that *sGLT1 siRNA-3* significantly reduced the expression of *sGLT1* (Figure 2A). In addition, *GLUT2 siRNA-2* was the most effective segment for silencing *GLUT2* (Figure 2B). Furthermore, under our experimental conditions, RT-PCR analysis showed that the *sGLT1* mRNA levels were reduced by 85% in Caco-2 cells after exposure to *SGLT1 siRNA-3* (Figure 2A). Moreover, *GLUT2* mRNA levels were reduced by 75% in *GLUT2 siRNA-2*-treated cells (Figure 2B). Collectively, *SGLT1 siRNA-3* and *GLUT2 siRNA-2* were declared to be the optimum interference segments.

### 3.5. Western Blot Analysis

To further clarify the mechanism underlying the transmembrane transport of ANC, we examined the expressions of *sGLT1* and *GLUT2* in the presence of ANC, FA-g-CS, or both after siRNA transfection. On the one hand, the Western blot assay showed that, in both cell lines treated with nonrelated and scrambled *siRNA*, the expressions of *sGLT1* and *GLUT2* were significantly elevated upon treatment with ANC, FA-g-CS, or their combination compared with the blank (Figure 3A,B). On the other hand, in both cell lines treated with siRNA transfection, the expression levels of *sGLT1* and *GLUT2* in the ANC-treated group and the FA-g-CS-treated group were significantly increased compared with the blank control. In particular, the expression of *sGLT1* and *GLUT2* increased by up to 90% after treatment with ANC and FA-g-CS. These results suggest that both ANC and FA-g-CS upregulate the expressions of *sGLT1* and *GLUT2*.

### 3.6. Immunofluorescence Staining

Immunofluorescence staining was performed to investigate whether the levels of *sGLT1* and *GLUT2* were increased due to the incubation with FA-g-CS, ANC, or their combinations. The results of the immunofluorescence staining revealed that the protein levels of *sGLT1* (Figure 4A) and *GLUT2* (Figure 4B) were increased when Caco-2 cells were exposed to ANC or FA-g-CS compared with the control cells (Figure 4).

### 3.7. Molecular Docking of Chitosan and ANC on sGLT1 and GLUT2

In order to further evaluate and verify the binding stability of *sGLT1*, *GLUT2*, FA-g-CS, and ANC (Cyanidin-3-glucoside) and their interaction mechanisms, a molecular docking modeling analysis was carried out. The binding energy function was calculated using the formula used in the previous study [33]. Figure 5 shows the structures of FA-g-CS and ANC.

The binding energies of *sGLT1* and FA-g-CS+ANC, *GLUT2* and FA-g-CS+ANC, *sGLT1* and ANC, and *GLUT2* and ANC were −13.85, −14.88, −7.16, and −7.60 kcal/mol, respectively, which can be categorized into two conventional combination types (hydrogen bonds and hydrophobic interactions). Molecular docking results show that FA-g-CS+ANC, ANC, *sGLT1*, and *GLUT2* have good binding effects and high matching degrees (binding energy below −6 kcal/mol). The hydroxyl groups of ANCs can form strong hydrogen bond interactions with *sGLT1* at the site of CYS-345, ARG-336, GLY-509, and ASP-273 with short hydrogen bond distance and strong binding force (Figure 6(1)). There are hydrogen bond interactions between ANC and *GLUT2* at the active sites of ASN-32, ASN-320, Trp-420, etc. (Figure 6(2)). Moreover, small molecules are inserted into the deep cavity of proteins, which means that small molecules are compatible with proteins and are conducive to forming stable complexes (Figure 6(2)). In order to further study the binding ability of protein cavities with multiple ligands, we conducted further studies on the docking protein sites of FA-g-CS after ANC entered the protein cavity. According to the docking results, FA-g-CS could enter the complex sites of ANC-*GLUT2* and ANC-*sGLT1* (Figure 6(3,4)), and the compound could also bond with the protein sites (GLN-36, GLN-35, ASN-32) with multiple hydrogen bond interactions. Importantly, the sugar ring of FA-g-CS can also form hydrogen bond with the hydroxy group of ANC, suggesting that the binding of ANC to protein pocket would promote the stabilization of FA-g-CS. FA-g-CS forms hydrogen bond interactions with ARG-259, ASP-454, and ASP-273 of ANC-*sGLT1*; subsequently, hydrogen bond interactions with ANC-*sGLT1* are formed. In conclusion, both FA-g-CS+ANC and ANC can form stable complexes with *sGLT1* and *GLUT2*.

## 4. Discussion

It is widely known that ANCs are beneficial to human health due to their natural antioxidant activity [34,35]. Therefore, we believe that it is important to increase knowledge on the ways of increasing their bioavailability. Numerous in vitro and in vivo experiments have proven that the small intestine plays an important role in the absorption and utilization of ANCs [3,36]. Caco-2 cells provide a well-established model for studying intestinal absorption in vitro [11,12,27,37]. Whether in animal studies or clinical studies, the appropriate ANC dosage for interventions is worth studying.

Papp values are used to evaluate the absorption efficiency of the loaded drug in vivo. Previous studies have shown that the Papp values of complete drug absorption are greater than 10^−4^ cm/s, while the Papp values range from 10^−1^ cm/s to 10^−6^ cm/s with the drug absorption rate between 1% and 100%, and the Papp values of the drugs with absorption rates less than 1% are clearly smaller. This study showed that the absorption rate of ANC is very low (Papp < 10^−6^ cm/s), which is consistent with the previous study [15,38]. In addition, although Papp values increased slowly with the increase in ANC concentration, they gradually reached saturation as the concentration continued to increase, indicating that the transport of ANC in Caco-2 cell model was not completely dependent on ANC concentration; this indicated that the transport of ANC may be mediated by transporters. With FA-g-CS co-treatment, the transfer volume of anthocyanins from A to B increased, indicating that FA-g-CS promoted the absorption of ANC.

The results of the MTT assay indicate that FA-g-CS+ANC has no toxic effect (with FA-g-CS concentration lower than 8.2 × 10^−4^ mg/mL; with ANC concentration lower than 2.06 × 10^−3^ mg/mL) on normal cells. It has been previously observed that different cytotoxicity assays lead to different results depending on the test reagents used and the cytotoxicity assay adopted [24]. According to the LDH assay, low concentrations of FA-g-CS and ANC, such as 9 × 10^−5^ and 2.3 × 10^−4^ mg/mL, respectively, are not toxic; however, as the concentration increases, the cytotoxicity will increase. These results match well with those from the MTT assay, although the qualified value from the MTT assay was bigger. Hence, 9 × 10^−5^ mg/mL FA-g-CS and 2.3 × 10^−4^ mg/mL ANC were employed for further experiments. As a result, ANCs encapsulated by FA-g-CS microcapsules are biocompatible, verifying that the availability of FA-g-CS to improve the bioavailability of ANCs is crucial, which requires further study.

Previous results showed that the cyanidin-3-O-β-glucoside was transported through the Caco-2 cell monolayers in intact glycone forms, mediated by transporters through passive transport into epithelial cells [11]. Candidates for ANC transporters were the glucose transporters. Among them, *sGLT1* and *GLUT2* were described widely in ANC transmembrane transport; moreover, a notable correlation was found between the relative absorption of different ANCs and their *GLUT2* and *sGLT1* molecular recognition [14]. The hypothesis raised is that ANCs could interfere with the transporters responsible for their own transport. Interactions between food components can change the conformation of ANCs and further influence the efficiency of transmembrane transport.

The present study examines the interactions between ANCs, FA-g-CS, and the transporters (*sGLT1* and *GLUT2*). Primarily, the transport of ANCs across Caco-2 cells incubated with 5 × 10^−5^ mg/mL FA-g-CS solution verified that FA-g-CS can promote ANC transmembrane transport. Subsequently, the *siRNA* results demonstrated that *sGLT1* and *GLUT2* participated in the transmembrane absorption of ANC, which supports previously described findings [39]. Simultaneous Western blot analyses consistently show elevated *sGLT1* and *GLUT2* protein expression in the presence of FA-g-CS, ANC, or their combination. These results suggest that FA-g-CS may modulate the efficiency of ANC transmembrane transport by increasing glucose transporter expression. It was previously found that FA treatment regulates *GLUT2* gene expression [19]. Results from the present study validate that *sGLT1* and *GLUT2* expression can be regulated by the activation or inhibition of one of multiple components [40]. Interestingly, our results reveal that cells incubated with both FA-g-CS and ANC significantly increased *sGLT1* and *GLUT2* levels compared with FA-g-CS and ANC alone. Meanwhile, it suggests that the interaction between FA-g-CS and ANC was substantial, thus warranting more attention [41]. This may be due to the variation of the ANC chemical structure caused by FA-g-CS [42]. Previous work has also suggested that polyphenols in sour cherries extracted by solid-phase extraction can decrease the transmembrane absorption efficiency of ANCs by fivefold–tenfold, which may be related to the interaction between ANCs and polyphenols [39].

Our immunofluorescence findings were consistent with the Western blot results. These findings further support the hypothesis that both the interactions between FA-g-CS and ANC and between FA-g-CS and transporters affect the transport of ANC.

The interactions between FA-g-CS, ANC, *sGLT1*, and *GLUT2* were evaluated through molecular docking modeling. The van der Waals forces between the small molecules and protein receptor residues significantly support the spatial conformation of the complex. Furthermore, ANC and FA-g-CS bound to *sGLT1* or *GLUT2* mainly through hydrophobic interactions and hydrogen bonds, forming stable complexes.

In summary, this work investigates the mechanisms of FA-g-CS—a self-made new hydroxycinnamoyl chitosan derivative—as a wall matrix to microencapsulate ANC on an ANC transmembrane transport. The results demonstrate that FA-g-CS is conducive to ANC transmembrane transport by regulating the expression of *sGLT1* and *GLUT2*. The FA-g-CS and ANC with *sGLT1*/*GLUT2* via hydrophobic forces and hydrogen bonds to finally form the FA-g-CS and ANC-*sGLT1*/*GLUT2* complexes, which increases the stability of ANC and provides a new means of improving the bioavailability of ANC.

## Figures and Tables

**Figure 1 foods-11-03299-f001:**
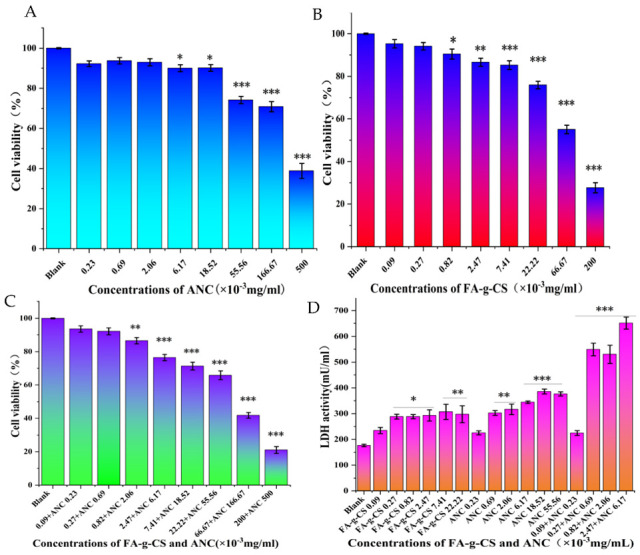
Cell viability of ANC (**A**), FA-g-CS (**B**), and FA-g-CS and ANC (**C**), as determined by MTT assay and LDH analysis (**D**). Asterisks indicate statistically significant differences from untreated cells (*—*p* < 0.05; **—*p* < 0.01; ***—*p* < 0.001).

**Figure 2 foods-11-03299-f002:**
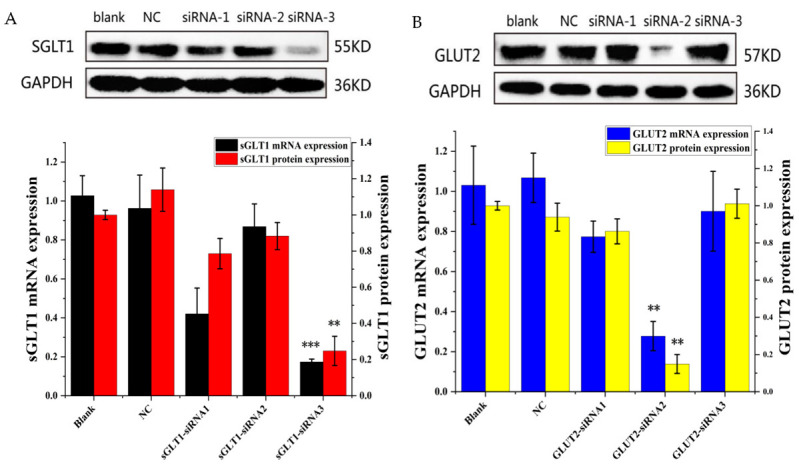
The expression levels of *sGLT1* (**A**) and *GLUT2* (**B**) in Caco-2 cells after *siRNA* transfection for 72 h, as detected by real-time PCR and Western blotting. blank—blank control; NC—negative control; **—*p* < 0.01 versus control; ***—*p* < 0.001 versus control.

**Figure 3 foods-11-03299-f003:**
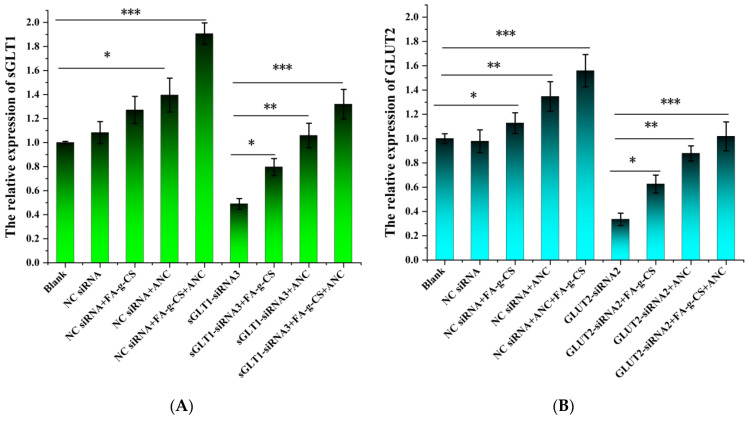
Expression of *sGLT1* and *GLUT2* in Caco-2 cells incubated with FA-g-CS, ANC, or their combination after siRNA transfection, as determined by Western blotting (**A**,**B**). *—*p* < 0.05 versus control; **—*p* < 0.01 versus control; ***—*p* < 0.001 versus control.

**Figure 4 foods-11-03299-f004:**
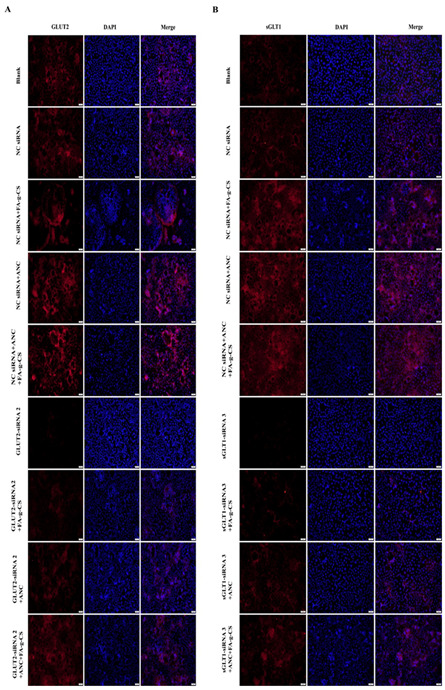
Effects of ANC and FA-g-CS on the protein levels of *GLUT2* (**A**) or *sGLT1* (**B**). Caco-2 cells were treated with the indicated ANC (2.3 × 10^−4^ mg/mL), FA-g-CS (9 × 10^−5^ mg/mL), or their combination for 48 h.

**Figure 5 foods-11-03299-f005:**
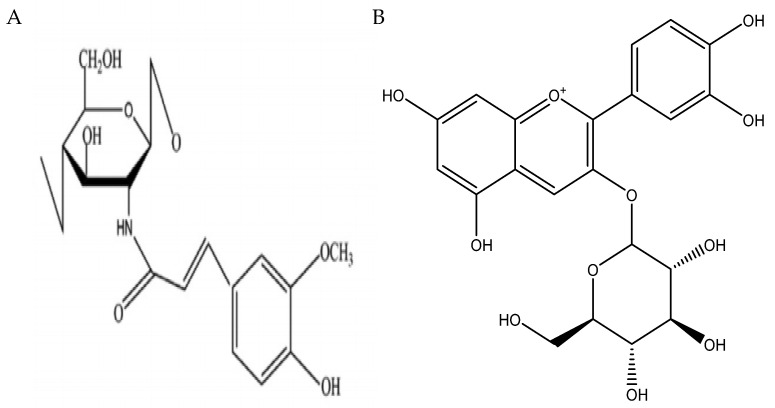
The structures of FA-g-CS (**A**) and ANC (**B**).

**Figure 6 foods-11-03299-f006:**
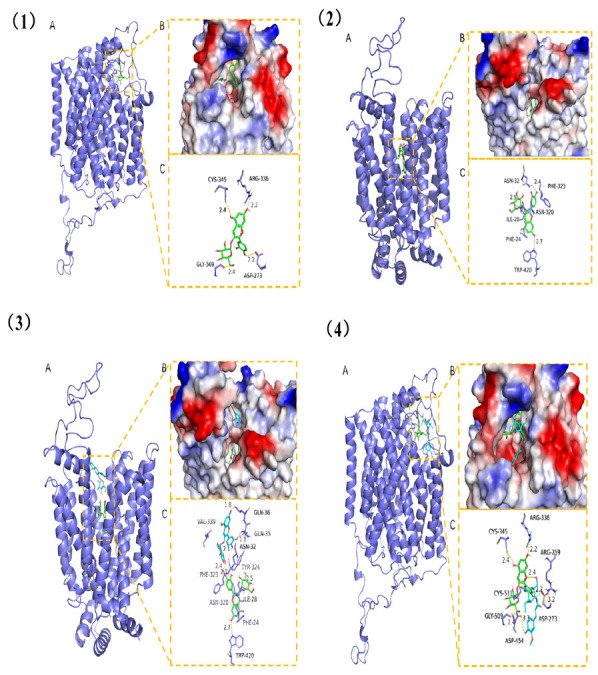
(**1**) The combination mode of ANC with *sGLT1*. (**A**) The 3D structure of ANC-*sGLT1* complex. (**B**) The surface detail of active site. (**C**) The detailed combination mode of ANC-*sGLT1* complex. The shape of the backbone of the protein is tubular and green in color. The compound is rendered by green. Yellow dash represents the hydrogen bond distance. (**2**) The combination mode of ANC with *GLUT2*. (**A**) The 3D structure of the complex. (**B**) The surface detail of the active site. (**C**) The detail binding mode of the ANC-*GLUT2* complex. The shape of the backbone of the protein is tubular and green in color. The compound is rendered by blue. Yellow dash represents the hydrogen bond distance. (**3**) The combination mode of ANC+FA-g-CS with *sGLT1*. (**A**) The 3D structure of the ANC+FA-g-CS-*sGLT1* complex. (**B**) The surface detail of the active site. (**C**) The detailed combination mode of the complex. The shape of the backbone of the protein is tubular and green in color. The compounds are rendered by green (ANC) and blue (FA-g-CS). Hydrogen bond distance is represented by a yellow dash. (**4**) The combination mode of ANC+FA-g-CS with *GLUT2*. (**A**) The 3D structure of the ANC+FA-g-CS-*GLUT2* complex. (**B**) The surface detail of the active site. (**C**) The detailed combination mode of the ANC+FA-g-CS-*GLUT2* complex. The shape of the backbone of the protein is tubular green in color. The compounds are rendered by green (ANC) and blue (FA-g-CS). The hydrogen bond distance is represented by a yellow dash.

**Table 1 foods-11-03299-t001:** Interfering RNA sequences.

Gene Name	RNA Sequence (5′-3′)
*NC siRNA*	UUCUCCGAACGUGUCACGUTT ACGUGACACGUUCGGAGAATT
*sGLT1 siRNA-1*	GGACAGTGTTGAACGTCAATT TTGACGTTCAACACGTACCTT
*sGLT1 siRNA-2*	GGGCCATATTCATCAATCTTT AGATTGATGAATATGGCCCTT
*sGLT1 siRNA-3*	GGAGCGTATTGACCTGGATTT ATCCAGGTCAATACGCTCCTT
*GLUT2 siRNA-1*	CAAACATTCTGTCATTAGTTT ACTAATGACAGAATGTTTGTT
*GLUT2 siRNA-2*	CGGGCATTCTTATTAGTCATT TGACTAATAAGAATGCCCGTT
*GLUT2 siRNA-3*	GTGCCATCTTCATGTCAGTTT ACTGACATGAAGATGGCACTT

**Table 2 foods-11-03299-t002:** PCR primer sequence.

Gene Name	RNA Sequence (5′-3′)	
*GAPDH*	Fw	TCAAGAAGGTGGTGAAGCAGG
	Rv	TCAAAGGTGGAGGAGTGGGT
*SGLT1*	Fw	GCAATCACTGCCCTTTAC
	Rv	TGTTGCCATCAGACACTATG
*GLUT2*	Fw	GCTACCGACAGCCTATTC
	Rv	AAACAAACATCCCACTCA

Fw—forward primer; Rv—reverse primer.

**Table 3 foods-11-03299-t003:** Transport parameters of ANC across Caco-2 cell monolayer at the timepoint of 4 h.

ANC Concentrations (×10^−3^ mg/mL)	Papp (×10^−7^ cm/s)		Papp (B→A)/ Papp (A→B)	Transport Efficiency A→B (%) without FA-g-CS	Transport Efficiency A→B (%) with FA-g-CS Co-Treatment
	B→A	A→B			
0.05	3.78 ± 0.48	4.91 ± 0.58	0.77	0.81	1
0.10	7.19 ± 0.98	9.61 ± 0.48	0.75	1.15	2.1
0.15	19.35 ± 9.08	15.02 ± 5.23	1.29	1.18	2.3
0.20	14.78 ± 0.48	11.20 ± 1.08	1.32	0.96	1.65

**Table 4 foods-11-03299-t004:** ANC transmembrane transport in 4 h across Caco-2 cell monolayer in presence or absence of FA-g-CS.

Concentrations of ANC at the Apical Side (×10^−3^ mg/mL)	Without FA-g-CS Co-Treatment, Concentrations of ANC at the Basolateral Side (×10^−3^ mg/mL)				In Presence of 0.05 × 10^−3^ mg/mL FA-g-CS, Concentrations of ANC at the Basolateral Side (×10^−3^ mg/mL)			
	1 h	2 h	3 h	4 h	1 h	2 h	3 h	4 h
0.05	0.0001 ± 0.00001 a	0.0003 ± 0.00001 b	0.0004 ± 0.00001 c	0.0004 ± 0.00001 c	0.0003 ± 0.00001 b	0.0004 ± 0.00001 c	0.0004 ± 0.00001 c	0.0005 ± 0.00001 d
0.10	0.0004 ± 0.00002 a	0.0008 ± 0.00001 b	0.0009 ± 0.00001 b	0.0012 ± 0.00002 c	0.0010 ± 0.0001 bc	0.0010 ± 0.0002 bc	0.0019 ± 0.0001 d	0.0021 ± 0.0002 d
0.15	0.0006 ± 0.00001 a	0.0009 ± 0.00002 b	0.0010 ± 0.0003 b	0.0018 ± 0.00001 c	0.0017 ± 0.0002 c	0.0025 ± 0.0002 d	0.0030 ± 0.0001 e	0.0035 ± 0.0001 f
0.20	0.0007 ± 0.00001 a	0.001 ± 0.0002 b	0.0012 ± 0.0001 b	0.0019 ± 0.0001 c	0.0020 ± 0.0001 c	0.0023 ± 0.0002 d	0.0032 ± 0.0001 e	0.0033 ± 0.0001 e

Different letters on the same line indicate significant differences.

## Data Availability

Data are included within the article.

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
