# Peer review of "The Effect of Ferulic Acid-Grafted Chitosan (FA-g-CS) on the Transmembrane Transport of Anthocyanins by *sGLT1* and *GLUT2"

_foods, 2022, doi:10.3390/foods11203299_

Round 1

Reviewer 1 Report

This work is devoted to the study of the effect of modified chitosan on transmembrane transport. The article is well written and clear. The relevance is beyond doubt. The following points I recommend to improve:

1. In the introduction, it is possible to update more exactly why such a study with such substances is the most important.

2. What is the error and accuracy of the experiment?

3. Please pay more attention to comparing the results with literature sources.

4. Please cite: 10.1016/j.molstruc.2021.131083.

5. Additional physicochemical research methods would be a good addition to this work.

Reviewer 2 Report

The manuscript by Ma et al aims to evaluate the effect of Fa-g-CS on transmembrane transport of anthocyanins. The manuscript sounds interesting, but has some flaws and needs to be significantly improved.

Specific comments.

1)      The abstract must be re-written, in order to clarify why siRNAs against sGLT1 and GLUT2 were used in Caco2 cells and which is the meaning of the results obtained. In the manuscript, Western blots are presented before immunofluorescence, claiming that immunofluorescence confirmed western blot data, but in the abstract the opposite was written. Furthermore, which immunofluorescence data? The results indicating that FA-g-CS enhance transmembrane transport should be better clarified.

2)      The introduction should cite reviews more than specific experimental works. For example, instead of references 1-5, general reviews should be cited. Ref. 8 should be better explained, instead of just pasting the title. The sentence in lines 47-49 is unclear. Line 64, why co-pigmentation? This seems and improper term in this context.

3)      Materials and methods. Fundamental information are lacking. For example, in paragraph 2.2 a description of Papp and how transport has been measured must be described. Furthemore, why the authors used mg/ml and why the specific specific concentrations were used (Lines 129-130)? These should be clarified here and in the Results section. Table 1: Orientation of the RNA sequence must be indicated. Is 5’-3’? Table 2: Forward and Reverse should be substituted with Fw and Rv and abbreviations detailed as table footnote. Paragraph 2.8: NC must be defined; Concentrations here are different from those indicated in the text. Why here molarity? Furthermore, details about primary and secondary antibodies used must be reported (which ones? Dilutions used? Cat. Number and company?). Paragraph 2.9: Which anthocyanin molecules have been used for molecular docking must be explicitly indicated here, in Results and Figure 5-Table 4.

4)      Results. Paragraph 3.1. Why is the analysis of transport reported before datat about cell viability and cytotoxicity. This is inconsistent. This paragraph should be reported after 3.2 and 3.3. Line 235. PDR(Papp(B→A)/ Papp(A→B)) must be defined and methods of measurement must be explained; Line 239. These data must be shown.

Figures with graphs are too tiny. Description of X and Y axis is not clearly visible and must be enlarged.

Paragraph 3.3: Panels A-C of Figure 1 must be cited here. Why were these concentrations chosen?

Lines 290 and 293. Blank and control group should be clearly indicated in the text.

Legend of Figure 3. NC should be defind.

Paragraph 3.6. If silenced, why are sGLT1 and GLUT2 still enhanced? Which is the meaning of this experiment?

5) Discussion. Lines 383-384: This sentence is not supported by data. These data must be necessarily shown. Lines 406-407. This sentence is also not supported by data shown. Only sGLT1 and GLUT2 mRNA and protein expression was analysed and not anthocyanin transport.

Reviewer 3 Report

Comments and recommendations to the authors

The question addressed by the paper is of interest for researchers in the field of polyphenols bioavailability and especially regarding anthocyanins which are known to have a low bioavailability. Proposing a potential solution to improve this absorption and demonstrating the mechanisms by which it can be improved is of importance.

The paper is very well written and complete in terms of experimental work performed to answer the questions raised by the subject covered. The appropriate experiments have been chosen and also the tests on cell viability and toxicity of the compounds are very important controls. They are very well described in the Material and Methods part, which is important for the reader to understand how scientific question has been addressed.

However, there is one major part which is really missing, knowing the topic of the paper. Data regarding ANC transport studies are reported in Table 3. However, the data obtained by the combination of FA-g-CS together with ANC are not. The effect of FA-g-CS on ANC transport t is the most important part of the paper. The data are not shown and only mentioned in a sentence (Lines 238-240). The authors mention that 0.05µg/mL FA-g-CS significantly increases the transport of ANC. The data should be presented for the combination of ANC and FA-g-CS as they are for the transport of ANC alone. Which concentration of ANC is affected by 0.05µg/mL FA-g-CS: 0.05, 0.10, 0.15 or 0.20µg/mL? Have the authors tested more concentrations of FA-g-CS? These missing data are really the most important part of the paper and should be shown in detail.

The following points are some comments and recommendations to the authors, proposed to improve the reading and understanding of the paper:

- In the 2.1. section, I would recommend adding the reference of the paper which describes how the FA-g-CS has been prepared (synthesis and characterization). I guess it has been prepared as described in the reference 23 but it could be mentioned in this section to be sure.

- In the 2.2. section, information regarding the way ANC and FA-g-CS have been solubilized prior to their addition to the buffer is missing. ANC commercial powder has probably been pre-dissolved in a solvent (which one?) before solubilization in buffer. What about FA-g-CS?

- Line 122: what are the conditions of the HPLC analysis mentioned? Should the authors refer at this place to the paper where they are described (reference 23)?

- Lines 182-186: for Western Blot assay, what is the time of incubation of the cells with ANC, with or without FA-g-CS, prior to protein extraction? According to line 160, Western Blot and PCR were done after 24, 48 and 72h following the siRNA treatment. Data of which time point are shown on Figure 2 and 3? Could the authors clarify in Material and methods or figures?

- General comment on the figures: the chart elements are difficult to read because very small, even on the online version of the paper. Could the authors magnify the legends?

- Line 271: the authors use the word “enzymes” to describe sGLT1 and GLUT2. “Transporters” would be more appropriate.

- Table 4 on page 13: The table does not bring more information than the lines 326-329 describing the results in the text. Moreover, the same structures draws are repeated on the different rows of the table. Either the table can be simplified or be completely avoided, as the data are clearly presented in few lines in the results section. Indeed, the structures of both ANC and FA-g-CS should be shown somewhere in the paper, in a separate figure for example.

- Line 383: the authors use the term “pre-treatment“ to qualify the addition of FA-g-CS to the cells and the effect on ANC increase of transport. If understood properly, FA-g-CS is added together with ANC on the cells, so it is not a per-treatment but rather a co-treatment. If FA-g-CS has been added to the cells a certain time before the transport study of ANC, it is not clear to the reader. That is also the reason why the transport study part with FA-g-CS should be as well described as the transport study of the compound ANC alone (in section 3.1 of the results and in section 2.2 of the material and methods).

- Lines 385-386: the data mentioned in these lines refer to Figure 1. The authors say that based on the MTT test, there is no toxic effect up to 0.82 µg/mL of FA-g-CS and up to 2.06 µg/mL for ANC. If we read the Figure 1, treatment starts to have a significant effect on cell viability at 6.17 µg/mL of ANC or 0.82 µg/mL for FA-g-CS. So, it seems that there is an inconsistency between the values of the figure and the text. Could the authors clarify which value is taken, the highest value having no significant toxic effect or the lowest value having a significant toxic effect?

Round 2

Reviewer 2 Report

The manuscript by Ma et al was improved, but the major flaw concerning data supporting that FA-g-CS increase the transmembrane transport of anthocyanins has not been properly addressed and the actual contribution of FA-g-CS cannot be appreciated (see below). Furthermore, some additional points need to be addressed. See below specific comments.

Specific comments.

1)      Abstract. Line 22. …”shows” is a wrong tense. Please, correct with “show”.

2)      Introduction. The authors improved references, but instead of Yang et al 2011 a more general review could be cited in the first sentence, such as 10.1111/1541-4337.12024

3)      Please, edit the manuscript. Some spaces are lacking (e.g. Line 14, 38 etc.)

4)      Materials and methods. Line 111. pH7.4

5)      Lines 129-130. Actually your previous study Ma et al 2016 has no indication about concentrations chosen here. You should better explain this point.

6)      Line 196. Two different secondary antibodies are indicated. Which one did you use? One of the two should be wrong.

7)      Results. Figures appear to be enlarged by stretching them. Font size of each column/histogram should be enlarged for all graphs and ug/ml should be replaced by mg/mL

8)      Line 255. Maybe better “up to 0.09 mg/mL FA-g-CS and 0.23 mg/mL ANC”?

9)      Paragraph 3.3 To how many hours of treatment do data presented in Table 3 correspond? 4 hours? Please, add this information in Table 3. This is also linked to the following comment.

10)   The comparison of transmembrane transport between free ANCs vs ANCs combined with FA-g-CS appears to be difficult to the reader, since Figure 2A and 2B have two different modes of presentation. Furthermore, free ANCs and ANCs combined with FA-g-CS are in two different graphs. My suggestion is that Figure 2B should be presented with the same column mode as in Figure 2A, with the same Y-axis and statistical analysis, in order to clearly compare the results presented in the two graphs. Furthermore, the authors should add these data in Table 3 (or in an additional Table identical to Table 3), in order to clearly compare the transport parameters of free ANCs and Fa-g-CS + ANCs. These data should be better shown, since they are the most important part of the manuscript. A description of statistics should be included in Figure2 legend.

11)   Paragraph 3.4 Here and throughout the manuscript, genes and siRNAs should be indicated in italics, whereas proteins not.
